# Unveiling the Moderating Factors in the Relationship between Physical Activity and Health-Related Quality of Life among University Students during COVID-19: A Mixed Study Design

**DOI:** 10.3390/healthcare12141389

**Published:** 2024-07-11

**Authors:** Eman M. Mortada, Nisreen N. Al Awaji, Uzma Zaidi, Monira I. Aldhahi, Hadel Alsaleh, Nouf Alroqaiba, Salwa S. Awad

**Affiliations:** 1Department of Health Sciences, College of Health and Rehabilitation Sciences, Princess Nourah bint Abdulrahman University (PNU), Riyadh 11671, Saudi Arabia; emmortada@pnu.edu.sa (E.M.M.); uazaidi@pnu.edu.sa (U.Z.); naalroqaiba@pnu.edu.sa (N.A.); 2Department of Health Communication Sciences, College of Health and Rehabilitation Sciences, Princess Nourah bint Abdulrahman University (PNU), Riyadh 11671, Saudi Arabia; nnalawaji@pnu.edu.sa (N.N.A.A.); hfalsaleh@pnu.edu.sa (H.A.); 3Department of Rehabilitation Sciences, College of Health and Rehabilitation Sciences, Princess Nourah bint Abdulrahman University (PNU), Riyadh 11671, Saudi Arabia; mialdhahi@pnu.edu.sa

**Keywords:** academic performance, health-related quality of life, health sciences university students, moderating effect, physical activity

## Abstract

This study examines the moderating effects of various factors on the relationship between physical activity (PA) and health-related quality of life (HRQoL) among 287 female college students during the COVID-19 pandemic. Data were collected through an online questionnaire covering health issues, PA, self-esteem, HRQoL, and sociodemographic information, supplemented by semi-structured interviews and focus groups with six participants. Results showed that 46% of respondents had good HRQoL, and 38% were physically active. The interactions of age, socioeconomic status (SES), and academic performance with PA on HRQoL were insignificant. However, physical and mental health problems significantly moderated the PA–HRQoL relationship, accounting for 1% and 4% of the variance, respectively. The qualitative analysis highlighted the need for student activity clubs and mental health support to enhance coping strategies and overall HRQoL. In conclusion, age and SES did not moderate the PA–HRQoL relationship, while physical and mental health issues did. Programs targeting students with health problems are crucial to improving their physical and mental health, thereby enhancing their well-being.

## 1. Introduction

The transition to university or college is a transformative period for students marked by increased autonomy, responsibilities, and academic and social pressures, significantly influencing students’ physical and mental health and the overall health-related quality of life (HRQoL). Challenges like maintaining a balanced diet, regular physical activity (PA), and consistent sleep patterns contribute to declines in physical health among university students, with first-year students often experiencing weight gain due to unhealthy eating habits and lack of exercise [1] alongside sleep deprivation, which can impair their academic performance [2]. Academic stress also influences physical activity levels and psychological well-being, collectively affecting HRQoL [3]. 

The COVID-19 pandemic has further exacerbated these challenges, leading to increased stress, changes in daily routines, and reduced opportunities for physical activity, which have negatively impacted students’ HRQoL [4]. Previous literature stresses the multiple challenges faced by young adults, negatively impacting their physical and mental health, thus compromising their HRQoL. These challenges include academic pressure, social integration, financial stress, and balancing personal and academic responsibilities [5,6]. Factors like sociodemographic elements, age, PA, and academic performance may moderate these effects on HRQoL [7,8]. Understanding these moderating factors is crucial for developing effective interventions.

PA is being increasingly studied for its numerous benefits to human health. Most of the massive amount of research on the topic strongly suggests that exercise and overall PA are associated with better quality of life and health outcomes [7]. More specifically, PA can ameliorate most of the physical and mental health issues commonly found among young adults. PA can help control body weight more effectively [9], improve sleep quality and prolong its duration [10], and may also improve the cognitive function of young adults [11]. In addition to its numerous benefits on physical health, PA has been found to improve the mental health status of young people [12] and improve their overall psychological well-being [13]. It has been shown to fight off depression in a massive study with more than 50,000 students aged 18–35 years.

Understanding how factors like age, socioeconomic status (SES), and academic performance moderate the relationship between PA and HRQoL is essential for developing tailored interventions. These interventions can maximize the health benefits of PA and improve HRQoL among college students. 

While it is well-documented that PA positively influences HRQoL among young adults, there remains a significant gap in understanding how individual factors like age, SES, and academic performance moderate this relationship, specifically in a college setting. This gap is crucial as these moderating factors can provide insights into developing more tailored and effective interventions. Therefore, this study seeks to explore these moderating effects using a mixed-methods approach, aiming to inform policies and interventions that can enhance the HRQoL of college students by addressing their unique needs.

The primary objective of this study is to investigate the moderating effects of age, sociodemographic factors, and academic performance on the relationship between PA and HRQoL in college students using a mixed-methods approach. Specifically, we aim to characterize PA and HRQoL patterns among students and to examine the effect of age, academic performance, socioeconomic status, and health status on the PA–HRQoL relationship. We hypothesized that PA may correlate with HRQoL and that this relationship is moderated by personal factors, including age, academic performance, socioeconomic status, and health status. The findings of this study are crucial for informing the design of interventions and policies aimed at promoting student health and well-being, thereby enhancing their HRQoL.

## 2. Materials and Methods

### 2.1. Study Design and Population

A cross-sectional study was conducted between January 2020 and August 2020 during the COVID-19 pandemic. To determine the required sample size, we made an assumption based on previous research [14], with a margin of error set at 5%, a z-score of 1.96, a 95% confidence level, and an effect size of 3. We estimated that a minimum of 287 participants (from a population of 762) was required for our sample [15]. We included junior college students aged between 18 and 21 years from Saudi Arabia who did not have any prior diagnosis of physical disabilities or a history of psychological problems. Participants must be non-smokers and must pass a medical screening survey to assess their overall health and confirm their eligibility. Participants must not have had any acute illness or injury requiring medical intervention in the past 6 months. The recruitment process employed a non-probability convenience sampling method, which resulted in the successful completion of the questionnaire by 287 individuals. We utilized the university’s email system to directly reach out to students and enhance recruitment efforts. Furthermore, an announcement about the study was posted in the media, and flyers were distributed among students to encourage participation in the survey. Moreover, a focus group was implemented to reduce any statistical biases and to understand the real concerns of the participants (*n* = 6).

### 2.2. Study Instruments/Measures

The questionnaire was administered online using Survey Monkey. The survey link was distributed to college students in Saudi Arabia. The participants were recruited through telephone or email invitations. The survey was designed with three main sections, which took approximately 15 min for participants to complete. The first section focused on sociodemographic and general health information from the respondents. It included questions related to age, sex, educational background, and health status. The second and third sections consisted of self-report questionnaires aimed to assess participants’ perceptions of their HRQoL and PA levels. These questionnaires were carefully selected to provide comprehensive insights into participants’ QoL and PA.

#### 2.2.1. Sociodemographic, Health, and Academic Performance Characteristics

Ten questions related to sociodemographic and general health information were administered to participants. Participants were requested to provide details regarding their age, nationality, gender, educational qualifications, and grade point average (GPA) during their academic years, including high school, bachelor’s degree, and postgraduate studies. Additionally, marital status, categorized as married or unmarried, and perceived economic status, classified as low, middle, or high income, were included in the survey. Furthermore, the questionnaire incorporated inquiries pertaining to the respondents’ general psychological health, distinguishing between the absence or presence of psychological illness, the existence or absence of disabilities, and their physical health status, differentiating between the presence or absence of health problems.

#### 2.2.2. Health-Related Quality of Life (HRQoL)

The HRQoL evaluation was conducted utilizing the Arabic version of the World Health Organization Quality of Life Questionnaire (WHOQOL-BREF), which has undergone prior validation and has been demonstrated to be reliable in the general Arab population [16]. This questionnaire comprises 26 items, which generated scores across five domains: general HRQoL and health status (2 items), physical health (7 items), psychological well-being (6 items), social relationships (3 items), and environmental health (8 items). A five-point Likert scale was used to rank each item, with higher scores indicating better quality of life. The Arabic version of the WHOQOL-BREF demonstrated significant results and excellent internal consistency (α = 95). To analyze the data, the three negatively framed items were reversed to positively framed items. Subsequently, the score for each domain was calculated, and all scores were multiplied by 4 to allow for comparison with scores derived from the WHOQOL-100. Raw scores were then converted into transformed scores using the transformation table provided in the manual. In the current study, the WHOQOL-BREF exhibited a favorable level of internal consistency, with a Cronbach’s alpha coefficient (α) of 0.86 

#### 2.2.3. Physical Activity Questionnaire

The PA level of the participants was assessed using a 7-day recall self-administered questionnaire known as the International Physical Activity Questionnaire-Short Form (IPAQ-SF) [17,18]. This questionnaire was designed to collect information about the frequency and duration of an individual’s PA over the past week. The IPAQ-SF categorizes PA into four main groups: vigorous, moderate, walking, and sitting. Participants were classified into three levels of PA: inactive, participants who reported no PA or engaged in insufficient activity to meet the criteria for minimally active or health-enhancing PA levels; minimally active, participants who engaged in three or more days of vigorous activity lasting at least 20 min per day, or five or more days of moderate-intensity activities, or walking for at least 30 min per day, or any combination of walking, moderate-intensity, or vigorous-intensity activities totaling at least 600 MET-min/week. The last category is active when participating in vigorous-intensity activities on at least three days, totaling at least 1500 MET-min/week, or engaged in seven or more days of any combination of walking, moderate-intensity, or vigorous-intensity activities, totaling at least 3000 MET-min/week. To classify the level of physical activity into ‘active’ and ‘inactive’, we used specific criteria based on guidelines from the World Health Organization. Active denotes participants engaging in at least 150 min of moderate-intensity or 75 min of high-intensity physical activity per week. Inactive indicates participants are not meeting the above criteria for physical activity [19].

The IPAQ-SF includes the calculation of metabolic equivalents (METs), which represent the energy expenditure associated with different types of activities. Consistent with established guidelines, IPAQ-related variables are reported in minutes per week (min/week) and METs (METs [min/week]) [20]. The validity of the IPAQ-SF was assessed using concurrent validity and criterion validity. Concurrent validity indicates reasonable agreement between the IPAQ-SF and the long-form questionnaire. In contrast, criterion validity demonstrates fair-to-moderate agreement between the IPAQ-SF and the Computer Science Application, Inc. accelerometer [16,17,21,22]. The IPAQ has a moderate ability to measure total weekly physical activity, as indicated by the correlation coefficients ranging from 0.71 to 0.89 [23]. 

#### 2.2.4. Focus Group

The qualitative approach was based on various grounded theories and provided information regarding the participants’ perceptions of HRQoL and PA during the focus group discussion. Discussions of focus groups were automatically recorded and transcribed, with the formal consent of the participants (*N* = 6 students). These qualitative methods allowed for a deeper exploration of barriers and attitudes of the students toward PA and HRQoL. An invitation was sent to the College of Health and Rehabilitation Sciences students and the heads of the departments for a focus group. Contact numbers and email addresses were provided during the invitation. Seven students voluntarily accepted the invitation to participate. Two psychologists, having a humanist approach from the research team, conducted a focus group to cover the participants’ bilingual issues. Six students participated in the focus group. Before the focus group discussion, the participants provided written consent. The purpose of the study and matters related to confidentiality and privacy were mentioned in the consent forms. However, the researchers verbally explained this information while opening the focus group. The major topics of the focus group discussions were related to PA and HRQoL. All digital audio recordings were electronically stored on a password-protected computer accessible only to the researchers. The voice-recorded data gathered from the focus group were then converted into transcripts. Furthermore, the data were assigned special identity numbers to keep the anonymity of the participants and divided into themes by experts. Major themes of COVID-19’s impact on PA and types of PA were discussed. For HRQoL, the major physical, psychological, social, and environmental aspects affected by the COVID-19 pandemic were explored. 

### 2.3. Ethical Considerations

The study was reviewed and approved by the Institutional Review Board of Princess Nourah bint Abdulrahman University (IRB: Log Number: 20-0191) in Riyadh, Saudi Arabia. At the beginning of the survey, the study’s description, informed consent, and confidentiality of information were obtained.

### 2.4. Statistical Analysis

Descriptive statistics were used to analyze the obtained data. For the qualitative variables, frequency and percentage were used, while for the quantitative variables, mean and standard deviation (SD) were used. The chi-square test was used to assess the association between the respondents’ personal characteristics, degree of HRQoL, and level of physical activity, in addition to using the odds ratio and confidence interval to measure the strength of association. Both the magnitude and direction of the correlations were investigated using Spearman’s rank correlation test. Potential moderators were added to six regression models using six centered interaction terms between PA and HRQoL to ascertain the moderating effects of the six covariates to minimize the potential multicollinearity and enhance interpretability on the relationship between physical activity as the independent variable and WHOHRQOL as the outcome variable. The interaction between each of the potential moderators’ age, SES, academic level, GPA, mental health, physical health, and physical activity was generated, resulting in 6 different interaction terms. Each interaction term was added separately to the model to test for potential moderation of each variable by physical activity on quality of life. The four assumptions of linearity, homoscedasticity, independence, and normality advocated by Osborne and Waters (2002) [24] were taken into consideration in order to ensure the statistical quality of the models that were presented. Statistical significance was set as ≤0.05. SPSS V20 was used for all analyses.

## 3. Results 

### 3.1. Quantitative Analysis

About 46% of the participants were categorized as having a good HRQoL (Table 1), and only 38.0% were classified as physically active (Table 2).

#### 3.1.1. Comparison of QoL Levels among the Respondent Students Based on Personal Characteristics

About 46% of the participants were categorized as having a good HRQoL (Table 1), and only 38.0% were classified as physically active (Table 2). The chi-square test was used to measure the association between the level of QoL among the respondent students and their personal characteristics. A statistically significant association was observed between the students’ age and their HRQoL (*p* < 0.001), with students older than 21 showing poor HRQoL (63.8%). Another significant association was between the level of HRQoL and SES (*p* = 0.017). In contrast, the majority of those with low SES (87.5%) showed poor HRQoL, with students who reported middle and high SES showing six and eight times significantly better HRQoL (OR: 6.39, 95% CI: 1.35–41.70 and OR: 8.08, 95% CI: 1.33–63.12, respectively). Academic level showed a significant association with the level of HRQoL (*p* = 0.02), as did the student GPA, with the students having a GPA of more than 4.5 exhibiting nine times better HRQoL (OR: 8.56, 95% CI: 4.8–15.1, *p* < 0.001). Moreover, there was a statistically significant association detected between the student’s level of HRQoL and the presence of both physical and mental health issues, as most of them (76.8 and 71.9%, respectively) had poor HRQoL (Table 1).

#### 3.1.2. Comparison of Physical Activity Levels among the Respondent Students According to Their Personal Characteristics

Regarding the relationship between the level of PA and the personal characteristics of the respondent students, the chi-square test revealed a statistically significant association between age group and GPA. Most students older than 21 years (68.8%) were physically inactive (*p* = 0.02). Conversely, the majority of students with a GPA above 4.5 (60.9%) were physically active and exhibited more than six times better quality of life (QoL) (OR: 6.44, 95% CI: 3.67–11.34, *p* < 0.001) (Table 2).

#### 3.1.3. Measuring the Relationship between the Level of Physical Activity and QoL

Despite the insignificant association between PA level and HRQoL (χ^2^ (1) = 1.19, *p* = 0.27), most of them (56.2%) were physically inactive and had a poor QoL (Figure 1).

#### 3.1.4. Correlation between Study Variables 

Table 3 displays Spearman’s correlation matrix of the study variables and reveals that HRQoL was significantly positively correlated with age, SES, and GPA (r = 0.05, 0.216, and 0.134, respectively; *p* < 0.01). PA was positively correlated with HRQoL (r = 0.058, *p* = 0.45), suggesting that the higher the level of PA, the higher the HRQoL.

However, a significant negative correlation was shown between the presence of both physical and mental health problems and WHO HRQoL (r = −0.091, −0.187, respectively, *p* < 0.01), indicating that the presence of health and mental issues lowers their WHO HRQoL. 

#### 3.1.5. Moderating Effects

The moderating effects of the six covariates on the relationship between PA and WHO HRQoL are displayed in Table 4, with WHO HRQoL as the dependent variable and the level of PA as the independent variable in all models. Potential moderators were added to each corresponding regression model using a centered interaction term. The interaction between each covariate and physical activity was generated, resulting in 6 different interaction terms. Each interaction term was added separately to the model to test for potential moderation of each variable by physical activity on quality of life.

The first regression model, age, is added to test for potential moderating effects. In this model, the influence of PA on HRQoL was insignificant (t = 0.93, β = 0.62, *p* = 0.36) but explained 7% of the WHO HRQoL variance (R^2^ = 0.07, F(2, 287) = 2.88, *p* = 0.43). The relationship between age and WHO HRQoL was significant (t = 1.09, *p* = 0.05).

In Table 4, the result reveals that age did not significantly moderate the relationship between PA level and HRQoL (*p* = 0.83). When SES was added to the second regression model to test its moderating effect, it showed a significant relationship with HRQoL (*p* < 0.00). The interaction between SES and PA for the HRQoL was insignificant (*p* = 0.39). This result indicates that SES did not moderate the relationship between the level of PA and HRQoL. 

Adding academic level to the third regression model and GPA to the fourth regression model showed significant relationships with HRQoL (*p* = 0.04 and *p* = 0.02, respectively). The interaction of academic level and GPA with PA to assess changes in HRQoL was insignificant (*p* = 0.28 and *p* = 0.23, respectively), suggesting that academic level and GPA did not have moderating effects on the relationship between the level of PA and HRQoL, accounting for only 1% and 2%, respectively, of the HRQoL variance (R^2^ = 0.01, F(2, 287) = 1.17, *p* = 0.28, and R^2^ = 0.02, F(2, 287) = 2.93, *p* = 0.49).

Testing the potential moderating effect of the presence of physical health problems in the fifth regression model and the presence of mental health problems in the sixth regression model showed significant relationships with the HRQoL (*p* < 0.01 and *p* < 0.01, respectively). The interaction of physical and mental health problems with PA in assessing the changes in HRQoL was significant (t = 2.69, β = 0.88, and t = 5.75, β = 0.87, *p* < 0.001, respectively), reflecting that the presence of physical and mental health problems had moderating effects on the relationship between the level of PA and HRQoL, accounting for 1% and 4%, respectively, of the explanatory power of HRQoL variance (R^2^ = 0.01, F(2, 287) = 8.22, and R^2^ = 0.04, F(2, 287) = 6.22).

### 3.2. Qualitative Analysis

Five questions were formulated for the focus group. The first question was related to physical activities/exercise. The next four questions were based on the QoL dimensions. Four main themes were identified regarding PA/exercise: PA during the pandemic, PA after the pandemic, type of activity, and reason for engagement/disengagement in physical activities and exercises. The theoretical framework of temporal self-regulation theory for exercise was used to identify the themes. Table 5 presents details of the main themes and subthemes.

#### 3.2.1. Physical Activity during the Pandemic

The analysis of the focus group interviews revealed two sub-themes for physical activity during the pandemic, namely active and passive. Some participants perceived the pandemic as a chance to take care of their physical health more actively. These respondents enjoyed exercising and were encouraged by their family members. However, those who had an exercise routine before the pandemic did not report any changes.

“… but during the pandemic, I used to be very energetic, and we started to walk in the neighborhood … in the house. My sisters and I cleared an area just for exercising”.(R2)

“… I completed my dancing classes”.(R4)

“… I put my yoga mat and began exercising because I had time. Then my mom said I’m so proud of you”.(R5)

“I didn’t notice the difference because I used to exercise at home”.(R3)

However, some of the respondents faced difficulty engaging in PA or exercise. Some attempted to join but could not maintain the routine.

“Before the pandemic, I was able to exercise”.(R1)

“… during the pandemic I heard about Cloe Ting and, then I actually start for two days then I come back to my old habits and eating”.(R6)

#### 3.2.2. Physical Activity after the Pandemic

The pandemic impacted human life and changed people’s perceptions. Respondents accepted that after the pandemic when they resumed normal life, their engagement with physical activities and exercise decreased or discontinued. Only those participants who were physically active, regardless of any change in circumstances, maintained their exercise routine. 

“I am actually walking too much before and after the pandemic. After the pandemic, I did exercise after every alternative day”.(R4)

“Before the pandemic, I was able to exercise for 30 min, e.g., walking. I tried to resume walking (after), but I found difficulty”.(R1)

“… when we came back to studying and everything (exercise) was just over and so I went back to my old habits surprisingly”.(R2)

“There was no time we had to study, to begin internship, and have so many things”.(R5)

#### 3.2.3. Types of Activities

The participants described a variety of activities, including walking, exercise, and dancing. However, some participants were unable to continue physical exercise.

“I thought if the COVID-19 pandemic didn’t happen, I wouldn’t know I loved exercise”.(R5)

“I don’t know, like dance classes, so I was very energetic, and I lost (loose) so much weight”.(R2)

“I completed my dancing class”.(R4)

“Attempted exercise but quit”.(R6)

#### 3.2.4. Reasons for Engagement/Disengagement in Physical Activities/Exercises

The focus group participants expressed various reasons for physical activities, including free time during the pandemic, accessibility to certain activities, and exercise as a habit. However, those who could not engage in physical exercise indicated environmental restrictions and sedentary lifestyles as reasons. 

“… but we stayed at home. Although mentally not OK stressful, it was a good time for me”.(R2)

“When the pandemic started, I think there was always time”.(R5)

“I tried to make an exercise with force, but I could not complete it so I stopped my exercises (but) I’m a dancer now”.(R4)

“I used to exercise at home”.(R3)

“… It’s actually difficult to change our behavior to fit in the environment”.(R1)

“I do not move, and I do not do anything, either exercise or walking, or I just eat and have fun”.(R6)

Regarding the dimensions of QoL, participants were asked the following question: What are the lessons learned related to the physical, psychological, and social health and environment that continued after the COVID-19 pandemic? Various theoretical frameworks were used, including Seligman’s helplessness, psychoanalysis, and coping mechanisms, to identify the themes. Table 6 shows the main themes and sub-themes of the four HRQoL dimensions.

The focus group analysis of HRQoL identified positive and negative physical lessons learned, with positive impacts, including pain acceptability, improved body image perception, increased awareness, and hygiene care. Negative impacts were identified as a lack of practicality and denial of physical health. 

“Accept to take the pain”.(R1)

“It helped me think … with dance; my face and body became very beautiful with dance. It helps me thinking meaning not only exercise”.(R4)

“No, before the pandemic, I did not know that all (yoga, candles, YouTube, Cloe Ting) but during the pandemic habits came out of nowhere”.(R5)

“Yes, a lot of things. When it comes to sanitizing, I still do that, yeah, because after the pandemic I felt like OK we live in a very toxic bacterial environment, it was just the fact that everything is disgusting and I became a little bit picky and sanitizing it”.(R2)

“I did not do any exercises (during pandemic), even though I know the benefits”.(R6)

“We don’t have time. Nothing to continue. I don’t think we have, actually”.(R3)

Regarding the psychological dimension of QoL, the qualitative analysis revealed two main themes of negative psychological effects manifested in psychological problems, and the second theme was related to coping mechanisms. 

“… By the way, I was diagnosed with insomnia. I’m not sure what the reasons were. This could be a distress. Perhaps I am ignoring myself. I take melatonin, but it doesn’t work”.(R1)

“I am active in classes but cannot sleep more than 2 h”.(R4)

“After the COVID-19 pandemic, I lost all my skills and abilities. I’m not sure about the outcome; I think something is wrong”.(R4)

“For me, I was stressful. Especially when I got the virus. I was 10 days in my room. I saw no one, and no one saw me, so it was difficult”.(R3)

“COVID-19 affected me in very bad or negative ways. I saw the whole world and everything is collapsing, and … actually you never know what’s going to happen if things go up and down the cases and trusted site”.(R2)

“I was crying all the time. I was sad. I have a lot of subjects to study. I lost many aspects of my personality. And this year, I think I lost myself. I even had doubts about my specialty … my relationships with the people, my friends. I was active on social media. … I closed the account”.(R6)

The focus group provided deep insights into the coping mechanisms they used to overcome or face the psychological impact of the pandemic. The major sub-themes of healthy coping were identified as the use of mindfulness, self-reinforcement statements, and here and now. By contrast, for unhealthy coping mechanisms, one participant expressed the use of regressive behavior, i.e., crying.

“The mindfulness, I use it, and it’s so helpful, and I’m still keeping using this technique and even the other technique, like taking a (deep) breathing”.(R1)

“… it was really stressful, but I tried to clear it out quickly”.(R2)

“We close that page (for the psychological effects of the pandemic). We moved forward”.(R5)

“Throughout the class, I feel I will lose some information; I will lose my attention. That never happened before. COVID. Oh my God, it has a negative effect on this point, on my mental skills. I am crying so much”.(R4)

Regarding the social health dimension of QoL, two themes of positive impact and negative impact were identified. 

“… after the pandemic, I felt like we are in a small world. We should get to know each other better because we are all humans. So now, right now, after the pandemic, I have so many friendships”.(R5)

“For me, I have become more social. I could not stay at home for a long time”.(R3)

“I think it made it better just a bit (with siblings) because they were very far and they eventually had to come together”.(R2)

“Previously, I was a social person. During the pandemic, I was stable, like giving support to my friends and our families, but after the pandemic, everything changed. Literally, I didn’t find any time for myself or any time to talk with anyone else”.(R1)

“I was an active person, and I have a lot of relationships. After COVID, I feel that I have lost my confidence”.(R4)

Regarding the environmental health dimension of HRQoL, two themes of positive impact and negative impact have been identified.

“I think that’s the best thing that’s happened in the environmental part that I tried”.(R2)

“After I became more anxious, I saw a lot of people in college, even in the hospital, my training, patients, and staff. It’s a lot because I used to see only my sister in my family, and suddenly, we are again back in life”.(R5)

“It’s like, you know, this is the issue I have to find a solution for”.(R1)

“I saw a lot of different levels of economic, social, and different people from different locations and real different economic, different thinking. I now feel during COVID, I was too stressed about how they (economically poor people) lived”.(R6)

## 4. Discussion 

### 4.1. HRQoL and Personal Characteristics

The study found a significant association between students’ age and HRQoL, with younger students (≤21) exhibiting better HRQoL than older students. Younger students may experience novelty and excitement with lighter workloads compared to those approaching graduation. University students, regardless of age, face higher stress-related issues compared to other age groups [25,26]. Female students tend to experience more stress but are more likely to use coping mechanisms than male students [8,27,28]. This study’s focus on female students highlights the need for future research to include males to verify the age–HRQoL relationship across genders. Additionally, students with lower SES have worse HRQoL due to financial strain and limited resources, consistent with previous research [29,30].

Higher grades and education levels were linked to better HRQoL. Middle academic level students (third year) had the lowest HRQoL, likely due to heavier workloads. Students with excellent GPAs (≥4.5 out of 5) had better HRQoL, likely due to increased self-esteem, goal accomplishment, and optimism, aligning with past research [31]. This may also relate to the sample size and methods used to assess HRQoL scores, indicating the need to consider these factors when interpreting results. This study also found a significant association between poor HRQoL and the presence of both physical and mental health issues, supported by [26,32].

### 4.2. Physical Activity and Personal Characteristics

This study found that older students (over 21) were less active than younger ones, consistent with other research showing a decline in physical activity with age among university students [33,34,35]. A study on Saudi adolescents found that physical activity decreases, especially among girls aged 13–17 [36]. This is due to adolescents’ shifting priorities and questioning academic or career paths [37]. These findings suggest tailored interventions are needed to encourage PA among older students, who are at higher risk of inactivity. The study also highlighted a significant association between SES and HRQoL, with 87.5% of low SES participants reporting poor HRQoL, while those from middle and high SES backgrounds perceived better HRQoL. Higher SES students often have greater access to resources, such as quality education, healthcare, housing, and more social support, which can buffer against stress and promote well-being. Low SES students may face chronic stress from financial insecurity, job instability, and discrimination, affecting their physical and mental health [38]. Other factors, like individual personality traits or health behaviors, could also play a role in the SES–HRQoL association.

The findings showed a significant association between both academic level and GPA with student HRQoL. Advanced coursework may demand better time management and study habits, developing transferable skills that contribute to higher HRQoL. Increased academic achievement can lead to self-esteem, confidence, and a sense of accomplishment [39]. 

### 4.3. Relationship between Physical Activity and Quality of Life

Regular exercise has been linked to improved HRQoL and better coping during COVID-19 [40,41]. Despite our findings, reverse causation or selection bias could explain the lack of significance. Regular PA is crucial for a healthy lifestyle, protecting against diseases and harmful behaviors, and enhancing HRQoL. However, PA engagement declines during adolescence and young adulthood, often due to increased academic demands [42,43,44].

Interestingly, students with higher GPAs were more likely to be physically active, suggesting a positive association between academic achievement and physical fitness. This highlights the importance of incorporating PA into students’ lives and exploring the potential reciprocal relationship between academic performance and an active lifestyle. 

While age, SES, academic level, and GPA did not moderate the PA–HRQoL relationship, mental and physical health issues did. Students with better mental and physical health showed a more pronounced positive impact of PA on HRQoL. Addressing these health issues is crucial for promoting overall well-being.

Qualitative findings supported the quantitative analysis with regard to the increased level of physical activity during the pandemic. However, many participants reduced or discontinued PA post-pandemic. Those who maintained active lifestyles before and after the pandemic emphasized personal healthcare styles. Marinating active lifestyles led to positive changes in body image, self-acceptance, and focus on health and functionality [45,46,47,48,49]. Increased hygiene practices became long-lasting habits [50]. The findings showed discrepancies between knowledge, attitudes, and behavior, leading to misinformation, skepticism, and denialism. This led to resistance toward measures such as masking, social distancing, and vaccination, weakening efforts to control the virus spread [51]. This led to apprehension and a lack of understanding about vaccines, increased doubtfulness, and rejection, affecting individual and public safety [52], which increased anxiety, stress, and depression due to constant uncertainty and conflicting information [53]. Denialism also causes delays in seeking health care services and the development of unsafe behaviors, worsening health outcomes [54]. Holding different attitudes can lead to social isolation and loneliness, particularly during a pandemic that demands social distancing [55]. 

### 4.4. Limitations of the Study

While this study covered multiple variables, it is important to acknowledge certain limitations. One such limitation is the potential influence of the fear of COVID-19 on participants’ perceptions and responses. Future post-pandemic studies could explore this variable in more detail to better understand its impact on various outcomes.

Moreover, the design of this study is observational and does not establish causation between variables. Further research using experimental designs or longitudinal studies is needed to explore causal relationships. Furthermore, the study’s sample predominantly comprises young and female individuals, which may not adequately represent the entire population. This limitation restricts the generalizability of the findings and does not provide insights into potential gender disparities or differences across age groups. The use of self-reported questionnaires in this study is also a limitation that may introduce social desirability bias, where respondents tend to provide answers that are socially acceptable or favorable rather than reflecting their true beliefs or behaviors. Future studies could consider using additional methods, such as observational data or objective measures, to mitigate this bias. Finally, in the qualitative part of the study, we did not create strata of students based on their academic performance. This approach may limit the depth of insights gained from different academic achievement levels and their potential impact on the studied variables.

### 4.5. Practical Applications

The findings of this study provide several practical implications for enhancing PA and overall quality of life among university students. Given the decline in PA among older students, universities should develop targeted programs, such as creating student activity clubs, organizing regular physical activity events, and providing accessible sports facilities [33,34,35]. Social elements like walking/running groups and incorporating enjoyable games and challenges can foster a love for movement. Flexible workout options, including online fitness classes, can cater to students with busy schedules [56].

Since mental health issues significantly moderate the PA–HRQoL relationship, universities should integrate mental health support services, such as regular check-ups, counseling, and stress management workshops, into their health programs [26,32]. Academic support programs, including tutoring, study groups, and time management workshops, can help students balance academic demands with physical and mental well-being [31,57]. 

Implementing comprehensive health education programs can raise awareness about maintaining a healthy lifestyle, covering topics like nutrition, exercise, and mental health [16,17]. Universities can run campaigns and workshops to promote balanced diets, regular exercise, and sufficient sleep [10,11]. PA programs should cater to diverse needs, including those who became active during the pandemic but discontinued afterward, ensuring inclusivity and adaptability to varying fitness levels [12,13]. Lastly, universities should create supportive environments with safe spaces for physical activity, healthy food options, and a community that values well-being [5,6]. By implementing these practical applications, universities can enhance the HRQoL of their students, promoting a healthier, more balanced, and more productive student life.

## 5. Conclusions

This study found that younger female university students have a higher quality of life, while socioeconomic status significantly impacts educational outcomes. Although higher GPAs correlate with better quality of life, this relationship needs further exploration. The decline in physical activity among older students highlights the need for targeted interventions. Regular physical activity has many benefits despite the fact that no direct link to quality of life has been found. The study recommends activating a physical activity student club and using modern tactics like social media to support physical activities. Addressing emotional and psychological issues with professional support is crucial. Holistic interventions are essential to improve students’ well-being, focusing on socioeconomic disparities, physical activity decline with age, and pandemic-related challenges.

## Figures and Tables

**Figure 1 healthcare-12-01389-f001:**
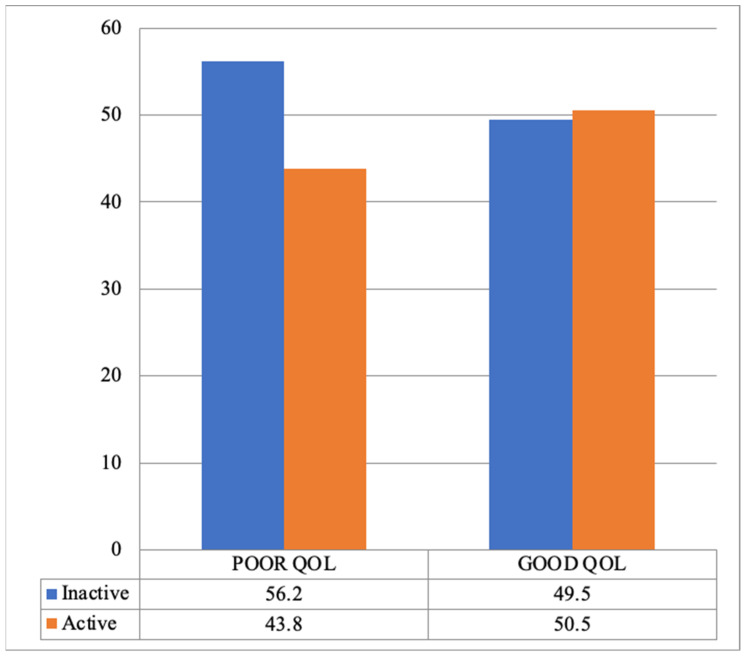
Assessment of the relationship between PA level and HRQoL.

**Table 1 healthcare-12-01389-t001:** Personal characteristics of the respondent students according to level of QoL (*N* = 287).

Characteristics	Responses	Overall	Level of QOL		X^2^ Test	*p*-Value	OR (95% CI)
		N (%)	Poor N(%)	Good N (%)			
**Age group**	**≤21**	149 (51.9)	66 (44.3)	83 (55.7)	10.93	<0.001 *	1
	**>21**	138 (48.1)	88 (63.8)	50 (36.2)			0.45 (0.27–0.75)
**Perceived SES**	**Low**	16 (5.6)	14 (87.5)	2 (12.5)	8.15	0.017 *	1
	**Middle**	243 (84.7)	127 (52.3)	116 (47.7)			6.39 (1.35–41.70)
	**High**	28 (9.8)	13 (46.4)	15 (53.6)			8.08 (1.33–63.12)
**Academic Level**	**4th**	83 (28.9)	33 (39.8)	50 (60.2)	9.63	0.022 *	1
	**6th**	95 (33.1)	59 (62.1)	36 (37.9)			0.40 (0.21–0.77)
	**8th**	100 (34.8)	57 (57.0)	43 (43.0)			0.50 (0.26–0.94)
	**12th**	9 (3.1)	5 (55.6)	4 (44.4)			0.50 (0.11–2.50)
**Department**	**Rehab**	78 (27.2)	44 (56.4)	34 (43.6)	7.24	0.065	1
	**Health Sciences**	121 (42.2)	71 (58.7)	50 (41.3)			0.91 (0.49–1.69)
	**Communication**	47 (16.4)	17 (36.2)	30 (63.8)			2.28 (1.02–5.16)
	**Radiology**	41 (14.3)	22 (53.7)	19 (46.3)			1.12 (0.49–2.56)
**GPA**	**≤4.5**	159 (55.4)	120 (75.4)	39 (24.5)	68.21	<0.001 *	1
	**>4.5**	128 (44.6)	34 (26.6)	94 (73.4)			8.56 (4.8–15.1)
**Health issues**	**Absent**	257 (89.5)	131 (51.0)	126 (49.0)	7.13	0.008 *	1
	**Present**	30 (10.5)	23 (76.7)	7 (23.3)			0.32 (0.12–0.81)
**Mental issues**	**Absent**	230 (80.1)	113 (49.1)	117 (50.9)	9.55	<0.001 *	1
	**Present**	57 (19.9)	41 (71.9)	16 (28.1)			0.38 (0.19–0.74)
**Total**		287 (100.0)	154 (53.7)	133 (46.3)			

* Significance difference (*p* ≤ 0.05); GPA: Students’ Grade Point Average; QOl: Quality of Life, SES: Socioeconomic Status; OR: Odds Ratio, CI: Confidence Interval. Data presented as Frequency (N) and percentage (%).

**Table 2 healthcare-12-01389-t002:** Comparing the personal characteristics of respondents with their pattern of physical activity (*N* = 287).

Characteristics	Responses	Overall N (%)	Level of Physical Activity		χ^2^ Test	*p*-Value	OR (95%CI)
			Inactive N (%)	Active N (%)			
**Age group (year)**	≤21	149 (51.9)	83 (55.7)	66 (44.3)	5.25	0.02 *	1
	>21	138 (48.1)	95 (68.8)	43 (31.2)			0.57 (0.35–0.92)
**Perceived SES**	Low	16 (5.6)	13 (81.2)	3 (18.8)	2.76	0.26	1
	Middle	243 (84.7)	148 (60.9)	95 (39.1)			2.78 (0.71–12.64)
	High	28 (9.8)	17 (60.7)	11 (39.3)			2.80 (0.55–15.96)
**Academic Level ****	4th	83 (28.9)	52 (62.7)	31 (37.3)	0.25	0.96	1
	6th	95 (33.1)	58 (61.1)	37 (38.9)			1.07 (0.56–2.05)
	8th	100 (34.8)	63 (63.0)	37 (37.0)			0.99 (0.52–1.88)
	12th	9 (3.1)	5 (55.6)	4 (44.4)			1.34 (0.28–6.37)
**Department**	Rehabilitation	78 (27.2)	51 (65.4)	27 (34.6)	1.626	0.89	1
	Health Sciences	121 (42.2)	74 (61.2)	47 (38.8)			1.20 (0.64–2.27)
	Communication	47 (16.4)	29 (61.7)	18 (38.3)			1.17 (0.52–2.66)
	Radiology	41 (14.3)	24 (58.5)	17 (41.5)			1.34 (0.57–3.13)
**GPA**	≤4.5	159 (55.4)	128 (80.5)	31 (19.5)	15.7	<0.001 *	1
	>4.5	128 (44.6)	50 (39.1)	78 (60.9)			6.44 (3.67–11.34)
**Health issues**	Absent	257 (89.5)	159 (61.9)	98 (38.1)	1.36	0.88	1
	Present	30 (10.5)	19 (63.3)	11 (36.7)			0.94 (0.429–2.06)
**Mental issues**	Absent	230 (80.1)	144 (62.6)	86 (37.4)	1.70	0.68	1
	Present	57 (19.9)	34 (59.6)	23 (40.4)			1.13 (0.63–2.49)
**Total**		287 (100.0)	178 (62.0)	109 (38.0)			

* Significance difference (*p* ≤ 0.05), ** fisher exact test was used; GPA: Students’ Grade Point Average; QOl: Quality of Life, SES: Socioeconomic Status; OR: Odds Ratio, CI: Confidence Interval.

**Table 3 healthcare-12-01389-t003:** Correlation between WHOQOL-BREF, physical activity, and personal characteristics among the respondent students (*N* = 287).

	1	2	3	4	5	6	7	8
**WHOQOL-BREF scale**	1							
**Physical** **activity**	0.065	1						
**Age**	0.051 *	−0.135 *	1					
**SES**	0.134 *	0.063	0.058	1				
**Academic Level**	0.064	−0.001	0.045	0.057	1			
**GPA**	−0.123 *	0.006	0.284 **	−0.057	−0.068	1		
**Health problems**	−0.158 **	−0.009	0.036	−0.007	0.028	0.070	1	
**Mental problem**	−0.182 **	0.024	−0.060	0.036	0.062	0.000	0.030	1

** Correlation is significant at the 0.01 level (2-tailed), * Correlation is significant at the 0.05 level (2-tailed). GPA: Students’ Grade Point Average; WHOQOL-BREF = World Health Organization Quality of Life-Brief.

**Table 4 healthcare-12-01389-t004:** Regression models illustrating the moderators’ effect on the relationships between physical activity and QOL.

MODEL		B	SE	β	T	*p*-Value	95% CI	
**1**	**Constant**	54.494	0.744		72.539	0.000	52.528	55.459
	**Moderator:** age	0.101	1.075	0.006	1.094	0.05 *	0.014	2.217
	**Predictor:** physical activity	0.616	0.666	0.068	0.926	0.355	−0.694	1.927
	**Interaction:** Age × PA	0.204	0.958	0.016	0.213	0.831	0.682	2.091
	**Model statistics**	F = 2.88, R^2^ = 0.07, ΔR^2^ = 0.006, *p* = 0.428						
**2**	**Constant**	54.04	0.52		103.48	<0.01	53.01	55.07
	**Moderator:** SES	1.89	0.53	0.21	3.62	<0.01 *	0.86	2.93
	**Predictor:** physical activity	0.66	0.52	0.07	1.26	0.21	−0.37	1.69
	**Interaction:** SES × PA	−0.44	0.51	−0.05	−0.86	0.39	−1.43	0.56
	**Model statistics**	F = 7.75, R^2^ = 0.05, ΔR^2^ = 0.002, *p* = 0.39						
**3**	**Constant**	54.23	0.68		79.98	<0.01	52.89	55.57
	**Moderator:** academic level	0.21	0.69	0.024	0.31	0.04 *	1.16	1.59
	**Predictor:** physical activity	−0.53	1.10	−0.28	−0.48	0.63	−2.69	1.64
	**Interaction:** academic level × PA	1.18	1.09	0.083	1.08	0.28	−0.96	3.32
	**Model statistics**	F = 1.17, R^2^ = 0.01, ΔR^2^ = 0.004, *p* = 0.28						
**4**	**Constant**	54.04	0.53		101.78	<0.01	52.99	55.083
	**Moderator:** GPA	1.29	0.56	0.14	2.32	0.021 *	0.19	2.39
	**Predictor:** physical activity	0.65	0.53	0.07	1.21	0.23	−0.40	1.69
	**Interaction:** GPA × PA	−0.34	0.46	−0.04	−0.73	0.47	−1.25	0.57
	**Model statistics**	F = 2.93, R^2^ = 0.024, ΔR^2^ = 0.03, *p* = 0.49						
**5**	**Constant**	54.47	1.30		38.78	<0.01 *	47.92	53.03
	**Moderator:** Physical health	1.52	0.25	−0.51	−1.09	<0.01 *	−1.63	−1.69
	**Predictor:** physical activity	0.26	0.35	0.03	0.739	0.461	−0.427	0.94
	**Interaction:** physical health × PA	0.29	0.62	0.88	2.694	<0.01 *	1.547	4.98
	**Model statistics**	F = 8.22, R^2^ = 0.012, ΔR^2^ = 0.005, *p* < 0.01						
**6**	**Constant**	54.89	0.56		98.79	<0.01 *	53.79	55.98
	**Moderator:** mental health	1.58	0.42	−0.99	−6.61	<0.01 *	−2.31	−15.86
	**Predictor:** physical activity	0.63	0.49	0.07	1.26	0.21	−0.36	1.61
	**Interaction:** mental health × PA	0.28	0.49	0.87	5.75	<0.01 *	3.39	19.18
	F = 6.22, R^2^ = 0.04, ΔR^2^ = 0.042, *p* = 0.002 *							

* *p* ≤ 0.05 is significance; B: unstandardized beta “regression coefficient”; SE: standard errors; β: standardized beta; F: ANOVA; ΔR^2^: R^2^ change; R^2^: coefficient of determination; 95% CI denotes 95% confidential interval (lower bound, upper bound), PA: physical activity, SES: socioeconomic status.

**Table 5 healthcare-12-01389-t005:** Description of PA themes.

	Themes of PA	Sub-Themes	Description	Example/Main Points
1	Physical activity during pandemic	Active	The status of physical activity during the pandemic	“…then I put my yoga mat and began exercising”.
		ii. Passive		“I am a sedentary person”.
2	Physical activity after the pandemic	Active	The status of physical activity after the pandemic	“…walking too much before and after the pandemic”.
		ii. Passive		“Then after that when we came back to studying and everything was just over”.
3	Type of Activity	Exercise	The nature of physical activities and exercises	“I wouldn’t know I would love exercise”.
		ii. Dance		“I complete my dancing class”.
		iii. Attempt to exercise for a short duration		“Attempted exercise but quit”.
4	Reasons for engagement/disengagement in physical activities/exercises	Free time	The motive for initiation of physical exercise	“When the pandemic started, I think there was time”.
		ii. Accessibility		“…could not have exercise materials at home, so I stopped my exercises (but) I’m a dancer now”.
		iii. Habitual		“I used to make exercise at home”.
		iv. Environmental restriction	The cause of stopping physical activities	“So difficult to try to change our activities to adapt to our environment”.
		v. Sedentary lifestyle		“I just take and eat and have fun”.

**Table 6 healthcare-12-01389-t006:** Description of QoL themes.

**A**	**Themes of Physical-QoL**	**Sub-Themes**	**Description**	**Example/Main Points**
1	Positive Impact	Acceptability of pain	Learning to live life despite pain	“Accept to take the pain”.
		ii. Improved body image	The perception of positive perception to own one’s body	“…my face and body become very beautiful with dance… It helps me thinking not only exercise”.
		iii. Increased awareness	To know about one’s own physical health	“…but during the pandemic habits came out of nowhere”.
		iv. Taking care of hygiene	To take responsibility for cleanliness	“When it comes to sanitizing, I still do that”.
2	Negative Impact	Lack of practicality	The discrepancy between knowledge and behavior	“…walking too much before and after the pandemic”.
		ii. Denial	A state where a person is unable to accept their reality	“Then after that when we came back to studying and everything was just over”.
**B**	**Themes of Psychological-QoL**	**Sub-themes**	**Description**	**Example/Main points**
1	Negative Psychological effects manifested in psychological problems	i. Insomnia	The mental status determines well-being or mental health issue	“I diagnosed with insomnia”.
		ii. Depression		“I lose all my skills and ability”.
		iii. Stress		“For me I was stressful”.
2	Coping Mechanism	Healthy coping Mindfulness Here and now	The adjustment style makes the adaptability easy	“I use mindfulness…breathing, other techniques”.
				“OK it was really stressful, but I tried to clear it out quickly”.
		ii. Unhealthy coping Regression	Behaviors not helpful for adjustment	“I am crying like so much”.
**C**	**Themes of Social-QoL**	**Sub-themes**	**Description**	**Example/Main points**
		i. Positive impact	Betterment in social relationships with others	“I become more social”.
		ii. Negative impact	Social relationships turn worse	“…but after the pandemic everything has changed, even my communication with my friends”.
**D**	**Themes of Environmental-QoL**	**Sub-themes**	**Description**	**Example/Main points**
		Positive impact	Positive outcome of resuming the normal environment after the pandemic	“I got to experience here in the practical part dealing with patients how to talk to them how to communicate with doctors as well as… the best thing that’s happened in the environmental part that I tried”.
		ii. Negative impact	Negative influence of resuming the normal environment after the pandemic	“I now feel during corona I was too stressed up so how they (lower socioeconomic status people) lived”.

## Data Availability

Data are contained within the article.

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
