# Peer review of "Unveiling the Moderating Factors in the Relationship between Physical Activity and Health-Related Quality of Life among University Students during COVID-19: A Mixed Study Design"

_healthcare, 2024, doi:10.3390/healthcare12141389_

Round 1

Reviewer 1 Report

Comments and Suggestions for Authors

It is extremely important to include the term social isolation or Covid-19 in the title;

I recommend removing the graph since it has no statistical relevance;

The discussion is too long and has unnecessary paragraphs;

The conclusion must be more summarized and respond to the objective of the work.

Author Response

Dear Reviewer 1,

Thank you for your time and the opportunity to revise our paper on “Unveiling the moderating factors in the relationship between physical activity and health-related quality of life among university students during COVID-19: A mixed study design”. The suggestions you offered have been constructive, and we also appreciate your insightful comments on revising the different sections of the paper.

We have considered and responded to the comments individually, indicating exactly how we addressed each concern or problem and describing the changes we have made. 

Comments of Reviewer 1

Authors’ Response

It is extremely important to include the term social isolation or Covid-19 in the title;

We have rectified the title to include the statement on COVID-19.

I recommend removing the graph since it has no statistical relevance;

Based on your valuable recommendations, it is removed, and modifications are made accordingly

The discussion is too long and has unnecessary paragraphs;

The discussion section is revised and shortened.

The conclusion must be more summarized and respond to the objective of the work.

The conclusion is now shortened with more focused information.

Please see the attachment. Changes are highlighted in yellow in the manuscript.

Reviewer 2 Report

Comments and Suggestions for Authors

Abstract

The abstract should not exceed 200 words in total.

Introduction:

-The introduction section clearly states the problem addressed by the article. Why is this article important? However, there should be a paragraph relating what is known so far about the objective of the study. In other words, a paragraph should be created in which it is shown how the variables being studied have been used by other authors.

Method:

-Add a flow chart according to PRISMA.

- I recommend following the SRQR (Reporting of qualitative research studies) see O'Brien et al. (2014):

O'Brien BC, Harris IB, Beckman TJ, Reed DA, Cook DA. Standards for reporting qualitative research: a synthesis of recommendations. Acad Med. 2014;89(9):1245-1251.

Results

Tables should follow the journal format.

Discussion 

-A title pointing out the limitations of the study should appear as a sub-section within the discussion (see 4.4).

- A Practical Applications section should appear at the end of the discussion 4.5). In it you should show how to apply the findings of this study to practical reality. You can include references, I attach an example that you can cite:

Albisua  Kaperotxipi,  N.  &  Zulaika  Isasti,  L.  M.  (2023). Influence  of  biographical  data  and  sports experience  of  university  students  of  physical-sports  education  on  their  beliefs,  attitudes  and physical self-concept. Espiral. Cuadernos del Profesorado, 16(32), 1-17.https://doi.org/10.25115/ecp.v16i32.7290

Conclusions

The conclusion section should be no longer than 10 lines. It should present the most relevant conclusions of the study.

References

The section on references and citations in the text needs a major overhaul.  In the link you can see the Guide on how to cite correctly (https://www.mdpi.com/authors/references) following the rules of the journal. In addition, I recommend updating and reducing the number of references.

Author Response

Dear Reviewer, 2,

Thank you very much for your valuable suggestions and comments on our manuscript. Those comments are of great assistance to us in improving and revising our manuscript. We have studied the comments carefully and have made corrections in line with the suggestions made by you. Revised sections are marked in yellow in the manuscript.

Below we provide the point-by-point responses from authors to your comments.

Comments of Reviewer 2

Authors’ Response

The abstract should not exceed 200 words in total.

The abstract is now within the 200-word limit

Introduction:

-The introduction section clearly states the problem addressed by the article. Why is this article important? However, there should be a paragraph relating what is known so far about the objective of the study. In other words, a paragraph should be created in which it is shown how the variables being studied have been used by other authors.

Thank you for your insightful comments. The introduction has been revised and the objective is now clearly stated.

Method:

-Add a flow chart according to PRISMA.

- I recommend following the SRQR (Reporting of qualitative research studies) see O'Brien et al. (2014):

O'Brien BC, Harris IB, Beckman TJ, Reed DA, Cook DA. Standards for reporting qualitative research: a synthesis of recommendations. Acad Med. 2014;89(9):1245-1251.

Changes as per your comments were made to the results

The response rate was 100.0%

Thanks for the article. It helped to address the overlooked issues. The recommended SRQR has been applied.

Tables should follow the journal format.

Thank you for this comment. Tables have been changed to meet the journal format.

Discussion

-A title pointing out the limitations of the study should appear as a sub-section within the discussion (see 4.4).

- A Practical Applications section should appear at the end of the discussion 4.5). In it you should show how to apply the findings of this study to practical reality. You can include references, I attach an example that you can cite:

Albisua  Kaperotxipi,  N.  &  Zulaika  Isasti,  L.  M.  (2023). Influence  of  biographical  data  and  sports experience  of  university  students  of  physical-sports  education  on  their  beliefs,  attitudes  and physical self-concept. Espiral. Cuadernos del Profesorado, 16(32), 1-17. https://doi.org/10.25115/ecp.v16i32.7290

The limitation of the study is now a sub-section within the discussion.

We added a section entitled Practical Applications that included recommendations of how to create a conducive university environment and programs. This section is supported by references.

Conclusions

The conclusion section should be no longer than 10 lines. It should present the most relevant conclusions of the study.

The conclusion is now shortened and more focused on the most relevant conclusions. 

References

The section on references and citations in the text needs a major overhaul.  In the link you can see the Guide on how to cite correctly (https://www.mdpi.com/authors/references) following the rules of the journal. In addition, I recommend updating and reducing the number of references.

The reference list and the intext citations are now amended according to the journal referencing style.  

Please see the attachment for the revised manuscript. 

Reviewer 3 Report

Comments and Suggestions for Authors

The introduction contains several sentences with no bibliographical support.

The authors mention variables in the objectives that they didn't mention in the introduction

The authors need to strengthen the gap in the literature that they intend to fill.

The hypotheses being studied should also be presented.

The authors mention that the study was carried out during the Covid pandemic, specifically in 2020. Four years on, why carry out this study when these relationships and the effect of Covid on the variables have already been studied extensively?

On the other hand, there is no mention of the pandemic in the introduction.

What are healthy students?

Why did they exclude people with disabilities?

The eligibility criteria need to be strengthened.

The authors need to better explain and present the study they used to calculate the sample size.

How long did it take to answer the questionnaires?

The regression models need to be presented in the statistical analysis. 

The results and discussion are very long and densely written, making the article difficult to read and follow.

Author Response

We appreciate your precious time in reviewing our paper and providing valuable comments. Your valuable and insightful comments led to possible improvements in the current version. The authors have carefully considered the comments and tried our best to address every one of them. We hope the manuscript after careful revisions meets your standards.

Below we provide the point-by-point responses. All modifications in the manuscript have been highlighted in yellow.

Comments of Reviewer 3

Authors’ Response

The introduction contains several sentences with no bibliographical support.

The introduction has been revised with the intention of being supported by references.

The authors mention variables in the objectives that they didn't mention in the introduction

The introduction is revised, and the objective is now clearly stated

The authors need to strengthen the gap in the literature that they intend to fill.

This has been mentioned in the introduction.

The hypotheses being studied should also be presented.

We have revised the introduction and added the statement of the hypothesis

The authors mention that the study was carried out during the Covid pandemic, specifically in 2020. Four years on, why carry out this study when these relationships and the effect of Covid on the variables have already been studied extensively?

While it is true that numerous studies have explored the relationships and effects of COVID-19 on various health variables, our research provides unique insights. The relationship between PA and HRQoL has been well established in the literature, highlighting the significant benefits of regular physical activity on overall health and well-being. The intricacies of this relationship are complex as they can be influenced by various multifaceted factors that can act as moderators. The three potential moderators that merit attention in this context are age, socioeconomic status (SES), and academic performance. Understanding how these factors moderate the relationship between PA and HRQoL is crucial for developing comprehensive interventions and policies that can better support health and well-being in both ongoing and future public health crises.

On the other hand, there is no mention of the pandemic in the introduction.

A phrase was added to the introduction.

What are healthy students?

Healthy students in this study are defined as those who are clear from any history of cardiometabolic, respiratory, and mental health issues. We screened participants through a detailed health history questionnaire to ensure that they did not have any underlying conditions that could affect the outcomes of the study.

Why did they exclude people with disabilities?

We excluded individuals with disabilities from this study to ensure the clarity and accuracy of our results. Including participants with disabilities could introduce additional factors and confounding variables related to physical activities, quality of life, and health status that are specific to their unique circumstances. These factors could complicate the analysis and potentially deviate the results from the true effects we aimed to measure. By focusing on a more homogenous group, we aimed to maintain the integrity of our findings and ensure that they are reliable and valid. Future studies could be designed specifically to explore the relationships in populations with disabilities to provide more tailored insights

The eligibility criteria need to be strengthened.

We have rectified and enhanced our inclusion criteria to ensure a more rigorous selection process.

The authors need to better explain and present the study they used to calculate the sample size.

Added study for sample size and reference.

Al Awaji, N., Zaidi, U,. Awad, S.S., Alroqaiba, N., Aldhahi, M.I., Alsaleh, H., Akil, S., Mortada,

E. M. (2022). Moderating Effects of Self-Esteem on the Relationship

between Communication Anxiety

and Academic Performance among

Female Health College Students

during the COVID-19 Pandemic. International journal of environmental research and public health, 19 (21),

13960. https://doi.org/10.3390/ijerph192113960

How long did it take to answer the questionnaires?

It took approximately 15 minutes for participants to complete the questionnaires. This duration has been added to the methods section to provide clarity on the time required from the participants.

The regression models need to be presented in the statistical analysis. 

The regression models are currently presented in the statistical analysis section.

The results and discussion are very long and densely written, making the article difficult to read and follow.

The results section of both the quantitative and qualitative sections have been revised as well as the discussion section.

Please see the attachment as it contains the revised manuscript 

Round 2

Reviewer 1 Report

Comments and Suggestions for Authors

ok

Author Response

Thank you very much.

Reviewer 2 Report

Comments and Suggestions for Authors

Authors' responses have helped to make the manuscript look very good. 

Author Response

Thank you very much.

Reviewer 3 Report

Comments and Suggestions for Authors

I would like to thank the authors for their effort in the revision. In my opinion, the results and discussion are still too long and densely written, making the article difficult to read and follow, which may diminish the interest of reading the article.

However, I think the article is fit for publication.

Author Response

Dear Reviewer 3,

The suggestions you offered have been constructive, and we also appreciate your insightful comments on revising the results and the discussion sections of the paper. We have carefully made the corrections as per your suggestions.

Below you will find responses from authors to your comments. The responses and the file were submitted last night. I hope you receive them this time. 

Comments of Reviewer 3 (Round 2)

Authors’ Response

I would like to thank the authors for their effort in the revision. In my opinion, the results and discussion are still too long and densely written, making the article difficult to read and follow, which may diminish the interest of reading the article.

We would like to express our gratitude for this valuable comment. To address this, we have employed the following changes:

The Results: We have carefully reviewed the results section and identified areas where we can present the findings more concisely. We summarized data, removed redundant details, and focused on the key results that support our conclusions.

The Discussion: We revised the discussion section to enhance clarity and flow. This was done by using clear transitions and focusing on the most significant implications of our findings to improve readability and follow-up.